# LGDiffGait: Local and Global Difference Learning for Gait Recognition with Silhouettes

## Abstract

The subtle differences between consecutive frames of a gait video sequence are crucial for accurate gait identification, as they reflect the distinctive movement of various body parts during an individual's walk. However, most existing methods often focus on capturing spatial-temporal features of entire gait sequences only, which results in the neglect of these nuances. To address the limitation, in this paper, we propose a new approach, named Local and Global Difference Learning for Gait Recognition with Silhouettes (LGDiffGait). Specifically, the differences within gait sequences are explicitly modeled at two levels: local window-level and global sequence-level. For the local window-level, we apply sliding windows along the temporal dimension to aggregate the window-level information, and the local movement is defined as the difference between pooled features of adjacent frames within each window. For the global sequence-level, global pooling across the entire sequence is employed, which is followed by subtraction to capture overall movement differences. Moreover, after difference feature learning, we develop a temporal alignment module to align these extracted local and global differences with the overall sequence dynamics, ensuring temporal consistency. By explicitly modeling these differences, LGDiffGait can capture the subtle movements of different body parts, enabling the extraction of more discriminative features. Our experimental results demonstrate that LGDiffGait achieves state-of-the-art performance on four publicly available datasets.

## 1 Introduction

Gait, defined as the way people walk, is a strong correlate of their identity Nambiar et al. (2019). Distinguished from other biometrics like face, iris, and fingerprint, gait is difficult to conceal and can be accurately recognized from a distance without requiring cooperation from the subject Sepas-Moghaddam & Etemad (2022). This unique advantage renders gait recognition highly suitable for various security applications, including suspect tracking, crime scene investigations, and identity verification, where long-distance and non-invasive identification is paramount Filipi Gonçalves dos Santos et al. (2022). However, the effectiveness of gait recognition can be significantly impacted by covariates (*e.g.*, clothing, carrying conditions, walking speed, and camera views), which pose challenges to maintaining high accuracy and robustness in real-world scenarios Li et al. (2020); Zou et al. (2024).

Currently, most existing gait recognition methods apply 2D convolutions to extract spatial features from an unordered set Chao et al. (2019); Hou et al. (2020; 2021); Fan et al. (2023b) or 3D convolutions to learn joint spatial-temporal representations from an ordered sequence Fan et al. (2020); Lin et al. (2021); Wang et al. (2023a); Fan et al. (2023a). The set-based approaches usually ignore the temporal order of input frames, treating them as independent entities, which can lead to the loss of important motion information that is crucial for accurate gait recognition. On the other hand, the sequence-based methods capture temporal dependencies using 3D CNNs, but they often fail to focus on fine-grained differences between frames Wang et al. (2023b).

Figure 1 shows an example of a gait silhouette sequence. As shown in Figure 1(a), the set-based approaches handle each frame independently, ignoring temporal continuity and causing significant

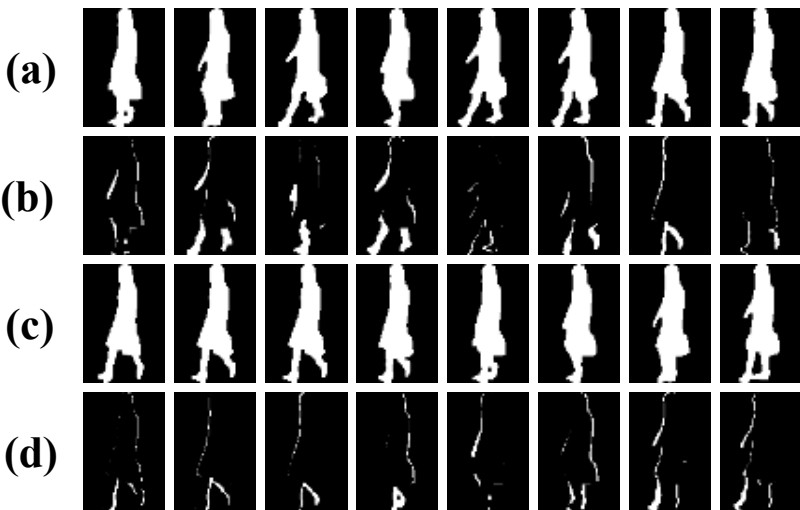

Figure 1: An example of a gait silhouette sequence. (a) shows silhouettes in an unordered set, while (b) presents the frame-level differences corresponding to (a). (c) depicts silhouettes arranged in an ordered sequence, and (d) illustrates the frame-level differences of (c). Due to the shorter time intervals between adjacent frames in the ordered sequence, the frame-level differences are more subtle.

differences between frames. This may result in missing the smooth transitions of leg swings and arm movements. Conversely, as depicted in Figure 1(c), sequence-based approaches maintain temporal dependencies, but the differences between adjacent frames are often very subtle. This subtlety poses challenges in capturing fine-grained movements.

To tackle the challenges of capturing subtle motion variations in gait recognition, we propose a novel framework called LGDiffGait. This framework integrates Local and Global Difference (LGDiff) blocks to learn discriminative gait representations. The LGDiff block consists of a Local Difference Module (LDM) and a Global Difference Module (GDM). The LDM aims to emphasize local movements and capture subtle variations between consecutive frames in local windows, ensuring that even minor movements are effectively captured. The GDM captures global differences by identifying significant deviations across the entire sequence that characterize individual gait patterns. To enhance the coherence and consistency of the captured motion information, a Temporal Alignment Module (TAM) is proposed after each difference module. This module aligns the extracted difference features with the sequence's overall dynamics, ensuring that the local and global differences are contextually relevant to the entire gait sequence.

The main contributions are summarized as follows:

- We propose LGDiffGait, a novel gait recognition framework that explicitly models local and global differences to capture comprehensive movement information.

- We propose the LGDiff block, which includes a Local Difference Module (LDM) and a Global Difference Module (GDM). The two modules work together to ensure detailed and holistic feature extraction, enhancing the LGDiffGait's ability to distinguish between subtle and significant gait variations.

- To ensure the extracted features are consistently aligned with the overall sequence dynamics, we propose a Temporal Alignment Module (TAM). This module enhances the coherence and integration of the local and global differences within the gait sequence, improving the robustness of the feature representations.

- Extensive experiments on the widely used gait datasets demonstrate the state-of-the-art performance of our method. Specifically, the mean Rank-1 accuracies achieved by our method are 95.2% on CASIA-B Yu et al. (2006), 92.3% on OUMVLP Takemura et al.

(2018), 82.7% on GREW Zhu et al. (2021), and 74.2% on Gait3D Zheng et al. (2022), respectively.

## 2 RELATED WORK

### 2.1 GAIT RECOGNITION

Gait recognition approaches can be roughly categorized into model-based and appearance-based methods. Model-based methods leverage the structural information of the human body to construct models that capture the kinematics of gait Zheng et al. (2022); Teepe et al. (2022); Cui & Kang (2023); Fu et al. (2023); Fan et al. (2024). These methods analyze the geometric and dynamic properties of the human body during walking, often requiring high-quality images for accurate pose estimation. Although model-based methods can provide robust recognition under varying conditions, they are computationally intensive and depend heavily on the quality of the input images Ma et al. (2023). Appearance-based methods analyze silhouettes of the gait cycle. Early works in appearance-based methods primarily used gait templates to represent gait patterns, such as the Gait Energy Image (GEI) Han & Bhanu (2005); Shiraga et al. (2016) and the Gait Flow Image (GFI) Lam et al. (2011). These templates summarize the motion characteristics of an individual's gait cycle into a single, static representation, simplifying the complexity of motion analysis. However, this simplification often comes at the cost of losing temporal and spatial details crucial for distinguishing between closely similar gait patterns. With the advent of deep learning, Convolutional Neural Networks (CNNs) have significantly advanced appearance-based gait recognition. Convolutional Neural Networks (CNNs) Fan et al. (2020); Chao et al. (2019); Lin et al. (2021); Fan et al. (2023b); Wang et al. (2023b) have advanced appearance-based methods by extracting robust spatial features, but most existing works can still overlook subtle motion details (*i.e.*, difference features) spread across multiple frames.

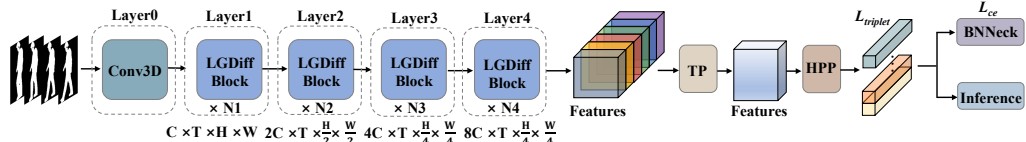

Figure 2: Overall framework of LGDiffGait. Each layer is composed of several LGDiff blocks. TP means Temporal Pooling. HPP is the Horizontal Pyramid Pooling.

### 2.2 GAIT DIFFERENCE MODELING

Differences in gait refer to the dynamic patterns of movement involved in human walking. Modeling these differences is crucial for recognizing unique walking patterns. Most existing works implicitly leverage these differences using 3D CNNs. Fan *et al.* Fan et al. (2020) propose Gait-Part, which captures local short-range spatio-temporal features using the Focal Convolution Layer (FCL) and Micro-motion Capture Module (MCM). Lin *et al.* Lin et al. (2021) present GaitGL, which employs a dual-branch structure within the Global and Local Convolutional Layer (GLCL) to extract global contextual information and local details. Huang *et al.* Huang et al. (2021) propose the Context-Sensitive Temporal Feature Learning (CSTL) network, which focuses on learning discriminative temporal representations from multi-scale temporal sequences. However, these methods primarily focus on spatial and temporal feature extraction without explicitly modeling frame differences, which can overlook subtle motion details critical for accurate gait recognition. A few studies explicitly model the dynamic gait features by focusing on the differences between frames. For example, Chen *et al.* Chen et al. (2009) develop the Frame Difference Energy Image (FDEI), which enhances dynamic feature modeling by integrating frame differences with dominant energy images derived from cluster-averaged silhouettes. Similarly, *et al.* Luo et al. (2015) propose the Accumulated Frame Difference Energy Image (AFDEI), which improves upon the Gait Energy Image (GEI) by incorporating temporal characteristics through the accumulation of frame differences, thereby enhancing the overall feature representation. However, both methods are template-based and may overlook finer temporal details. More recently, Wang *et al.* Wang et al. (2023b) propose DyGait,

which focuses on dynamic features using the Dynamic Augmentation Module (DAM). DAM generates a gait template from global temporal information and computes differences between each frame's feature maps and the template. While DyGait captures dynamic components, it primarily considers global average differences and may overlook local variations between adjacent frames.

In this paper, we propose LGDiffGait, which explicitly models gait differences using a Local Difference Module (LDM) to capture subtle frame-to-frame variations and a Global Difference Module (GDM) to identify significant sequence-wide deviations.

## 3 METHOD

### 3.1 FRAMEWORK OVERIVEW

The overall framework of the proposed method is shown in Figure 2, which consists of an initial convolutional block and four layers. Given a gait sequence input $G \in \mathbb{R}^{C_{in} \times T \times H \times W}$ with $C_{in}$ channels, $T$ frames, and $H \times W$ pixels, we first use a 3D convolution to extract shallow spatial-temporal features from each frame. The extracted feature $F_{in} \in \mathbb{R}^{C \times T \times H \times W}$ are then fed into the LGDiff blocks for local and global difference modeling. After that, we apply Temporal Pooling (TP) to aggregate temporal features. Finally, the aggregated features are fed into the Horizontal Pyramid Pooling (HPP) Fu et al. (2019) and the BNNeck Luo et al. (2019), with triplet loss Hermans et al. (2017) and cross-entropy loss to train the network.

### 3.2 LGDIFF BLOCK

The LGDiff block is a core component of the proposed framework, designed to model local and global differences within gait sequences. As can be seen in Figure 3, each block contains four sub-modules: the main convolutional module, local difference module, global difference module, and temporal alignment module.

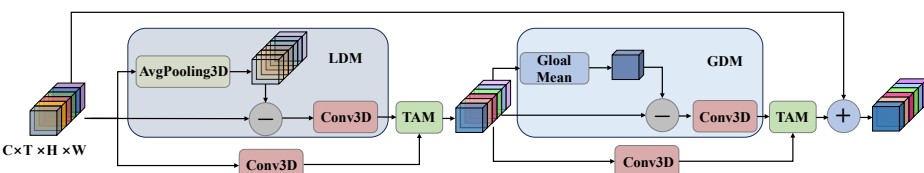

Figure 3: The architecture of LGDiff block, where $\ominus$ represents element-wise subtraction and $\oplus$ represents element-wise sum operation for residual connection.

#### 3.2.1 MAIN CONVOLUTIONAL MODULE

The main convolutional module captures spatial-temporal features across the entire sequence of gait frames. It starts with a 3D convolution operation that extracts motion patterns by simultaneously considering spatial and temporal dimensions. This is followed by batch normalization to stabilize and accelerate the training process, and a ReLU activation function to introduce non-linearity, enabling the model to learn more complex dynamics within the gait sequence. This process is formulated as:

$$F_{\text{out}} = \text{ReLU}(\text{BN}(\text{Conv}^{3 \times 3 \times 3}(F_{\text{in}})))$$ (1)

where $F_{\text{in}} \in \mathbb{R}^{C \times T \times H \times W}$ represents the input feature map with $C$ channels, $T$ frames, and spatial dimensions $H \times W$. $\text{Conv}^{3 \times 3 \times 3}$ is the 3D convolution with a kernel size of $3 \times 3 \times 3$. BN and ReLU are batch normalization and ReLU activation, respectively.

#### 3.2.2 LOCAL DIFFERENCE MODULE

The Local Difference Module (LDM) is designed to capture subtle variations between consecutive frames within a gait sequence, which are critical for accurate gait recognition. To achieve this, the

LDM uses a sliding window approach to calculate the differences between adjacent frames. Specifically, the input feature map is padded by replicating the first and last frames, allowing the sliding window pooling to operate across the entire sequence without changing the temporal length. The LDM then computes local differences by subtracting the average of frames within each sliding window from the current frame. This approach emphasizes short-range temporal variations, enabling the model to capture subtle and local movements. These local difference features can be denoted as:

$$F_{\text{l\_diff}} = F_{\text{in}} - \text{AvgPool3d}^{3 \times 1 \times 1}(F_{\text{in}}) \tag{2}$$

where $F_{\text{in}}$ represents the input feature map, while $\text{AvgPool3d}^{3 \times 1 \times 1}$ denotes the 3D average pooling operation with a kernel size and a stride of $3 \times 1 \times 1$ along the temporal dimension.

Once these local differences are computed, they are passed through a convolutional layer that shares the same architecture with the main convolutional module. This ensures that the local difference features are processed at the same level as the main features, benefiting the further integration of main dynamic features and local difference features.

### 3.2.3 GLOBAL DIFFERENCE MODULE

In contrast to the Local Difference Module (LDM), the Global Difference Module (GDM) is designed to capture long-range dependencies by computing differences across the entire sequence. The GDM operates by comparing each frame against a global representation of the sequence, which is obtained by averaging the features across all frames. This module enables the model to detect significant deviations from the overall gait pattern. This process can be described as:

$$F_{\text{g\_diff}}(t) = F_{\text{in}}(t) - \text{GlobalMean}(F_{\text{in}}) \tag{3}$$

where $F_{\text{in}}(t)$ represents the input feature map for frame $t$, and GlobalMean is the global average pooling along temporal dimension across all frames in the sequence. The resulting global difference features are then processed through a convolutional layer that mirrors the architecture of the main convolutional branches but does not share weights. These refined global difference features are subsequently merged with the main features, ensuring that the model captures both fine-grained and broad temporal dynamics across the entire gait sequence.

### 3.2.4 TEMPORAL ALIGNMENT MODULE

To ensure the coherence and consistency of the difference features with the overall sequence dynamics, Temporal Alignment Modules (TAMs) are deisned after each difference calculation. These modules consist of a sequence of convolutional operations that align the local and global difference features with the main features. Specifically, as depicted in Figure 4, the alignment is achieved by concatenating the main features with the difference features, followed by an MLP to reduce channels and a 3D convolutional operation that aligned the spatial-temporal information. Finally, a residual connection is added by summing the aligned features with the main features before applying an activation function. This process can be formulated as:

$$F_{\text{cat}} = \text{Conv}^{1 \times 1 \times 1}(\text{Cat}(F_{\text{main}}, F_{\text{diff}})) \tag{4}$$

$$F_{\text{aligned}} = \text{ReLU}(F_{\text{main}} + \text{BN}(\text{Conv}^{3 \times 3 \times 3}(F_{\text{cat}}))) \tag{5}$$

where $\text{Cat}(F_{\text{main}}, F_{\text{diff}})$ denotes the concatenation of the main features and the difference features on channel dimension, $\text{Conv}^{1 \times 1 \times 1}$ reduces the channel dimensionality, and $\text{Conv}^{3 \times 3 \times 3}$ aligns the features spatially and temporally. This step ensures that the final output from the LGDiff block is contextually relevant to the entire gait sequence, improving the integration of local and global difference features.

## 3.3 TRAINING LOSS

The training process of the proposed LGDiffGait framework is guided by a combination of triplet loss and cross-entropy loss, both of which are commonly used in biometric recognition tasks to enhance feature discrimination and classification accuracy.

Triplet loss is employed to increase the discriminative power of the learned features by ensuring that the distance between an anchor and a positive sample (same identity) is smaller than the distance

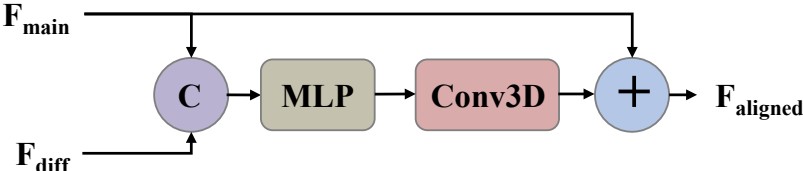

Figure 4: Diagram of Temporal Alignment Module (TAM). ⓒ means concatenation on the channels. The MLP is a Conv$^{1\times1\times1}$ layer to reduce channels.

between the anchor and a negative sample (different identity) by a specified margin. Given an anchor $\mathbf{x}_a$, a positive sample $\mathbf{x}_p$, and a negative sample $\mathbf{x}_n$, the triplet loss is defined as:

$$\mathcal{L}_{\text{triplet}} = \max(0, \|\mathbf{x}_a - \mathbf{x}_p\|^2 - \|\mathbf{x}_a - \mathbf{x}_n\|^2 + \alpha) \tag{6}$$

where $\alpha$ is the margin that separates the positive and negative pairs and is set to 0.2 by default. By minimizing this loss, the network is encouraged to learn feature embeddings that are closer for samples of the same identity and farther apart for samples of different identities.

Cross-entropy loss is applied to ensure accurate classification of the gait features into the correct identity classes. For a given input feature $\mathbf{z}$ and its corresponding class label $y$, the cross-entropy loss is defined as:

$$\mathcal{L}_{\text{CE}} = -\log\left(\frac{e^{\mathbf{w}_y^T \mathbf{z}}}{\sum_{k=1}^{K} e^{\mathbf{w}_k^T \mathbf{z}}}\right) \tag{7}$$

where $\mathbf{w}_y$ is the weight vector corresponding to the correct class $y$, and $K$ is the total number of classes. This loss function encourages the network to output a high probability for the correct class, thus improving the classification performance.

The final loss function used to train the LGDiffGait framework is a sum of the triplet loss and cross-entropy loss:

$$\mathcal{L} = \mathcal{L}_{\text{triplet}} + \mathcal{L}_{\text{CE}} \tag{8}$$

## 4 EXPERIMENTS

### 4.1 DATASETS

We have evaluated our LGDiffGait on four popular datasets, including CASIA-B Yu et al. (2006), OUMVLP Takemura et al. (2018), GREW Zhu et al. (2021), and Gait3D Zheng et al. (2022).

**CASIA-B** is a widely used gait dataset with 124 subjects, each captured from 11 views (0° to 180°) across 10 sequences under three conditions: normal (NM), carrying bags (BG), and wearing coats (CL). The first 74 subjects are for training, and the remaining 50 are for testing. During testing, the first four sequences (NM) serve as the gallery, and the rest serve as the probe set.

**OUMVLP** is one of the largest public gait datasets, containing 10,307 subjects, evenly split for training and testing. Each subject is recorded from 14 views (0° to 90°, 180° to 270°), with two sequences per view. For testing, sequences labeled "01" form the gallery, and "02" form the probe set.

**GREW** is a large-scale gait dataset for recognition in the wild, comprising 128,671 sequences from 26,345 individuals. The dataset is divided into 102,887 sequences for training and 24,000 for testing. In testing, each subject has two sequences in the gallery and two in the probe set.

**Gait3D** is an indoor gait dataset for uncontrolled environments, with 25,309 sequences from 4,000 subjects. The training set includes 18,940 sequences, while the testing set has 6,369 sequences. During testing, one sequence per subject is the probe, and the rest form the gallery. Evaluation metrics include accuracy, mean Average Precision (mAP), and mean Inverse Negative Penalty (mINP) Ye et al. (2021).

Table 1: Rank-1 accuracy (%) on CASIA-B Yu et al. (2006), and the identical-view cases are excluded.

| | Method | Venue | Probe View | | | | | | | | | | | Mean |
|---|---|---|---|---|---|---|---|---|---|---|---|---|---|---|
| | | | 0° | 18° | 36° | 54° | 72° | 90° | 108° | 126° | 144° | 162° | 180° | |
| NM | GaitSet Chao et al. (2019) | AAAI'19 | 90.8 | 97.9 | 99.4 | 96.9 | 93.6 | 91.7 | 95.0 | 97.8 | 98.9 | 96.8 | 85.8 | 95.0 |
| | GaitPart Fan et al. (2020) | CVPR'20 | 94.1 | 98.6 | 99.3 | 98.5 | 94.0 | 92.3 | 95.9 | 98.4 | 99.2 | 97.8 | 90.4 | 96.2 |
| | GaitGL Lin et al. (2021) | ICCV'21 | 96.0 | 98.3 | 99.0 | 97.9 | 96.9 | 95.4 | 97.0 | 98.9 | 99.3 | 98.8 | 94.0 | 97.4 |
| | CSTL Huang et al. (2021) | ICCV'21 | 97.2 | 99.0 | 99.2 | 99.2 | 98.1 | 96.2 | 95.5 | 97.7 | 98.7 | 99.2 | 96.5 | 97.8 |
| | DyGait Wang et al. (2023b) | ICCV'23 | 97.4 | 98.9 | 99.2 | 98.3 | **97.7** | 96.8 | 98.2 | 99.3 | 99.3 | 99.2 | 97.6 | 98.4 |
| | GaitBase Fan et al. (2023b) | CVPR'23 | - | - | - | - | - | - | - | - | - | - | - | 97.6 |
| | DANet Ma et al. (2023) | CVPR'23 | 96.4 | **99.1** | 99.2 | 98.2 | 96.6 | 95.5 | 97.6 | 99.4 | 99.5 | **99.3** | 96.9 | 98.0 |
| | GaitGCI Dou et al. (2023) | CVPR'23 | - | - | - | - | - | - | - | - | - | - | - | 98.4 |
| | QAGait Wang et al. (2024) | AAAI'24 | - | - | - | - | - | - | - | - | - | - | - | 97.9 |
| | CLASH Dou et al. (2024) | TIP'24 | - | - | - | - | - | - | - | - | - | - | - | 98.3 |
| | CLTD Xiong et al. (2024) | ECCV'24 | - | - | - | - | - | - | - | - | - | - | - | 98.6 |
| | LGDiffGait (Ours) | - | **98.4** | 98.7 | **99.6** | 98.9 | 97.1 | **97.6** | 99.3 | 99.8 | 99.6 | 98.9 | 98.5 | 98.8 |
| BG | GaitSet Chao et al. (2019) | AAAI'19 | 83.8 | 91.2 | 91.8 | 88.8 | 83.3 | 81.0 | 84.1 | 90.0 | 92.2 | 94.4 | 79.0 | 87.2 |
| | GaitPart Fan et al. (2020) | CVPR'20 | 89.1 | 94.8 | 96.7 | 95.1 | 88.3 | 84.9 | 89.0 | 93.5 | 96.1 | 93.8 | 85.8 | 91.5 |
| | GaitGL Lin et al. (2021) | ICCV'21 | 92.6 | 96.6 | 96.8 | 95.5 | 93.5 | 89.3 | 92.2 | 96.5 | 98.2 | 96.9 | 91.5 | 94.5 |
| | CSTL Huang et al. (2021) | ICCV'21 | 91.7 | 96.5 | 97.0 | 95.4 | 90.9 | 88.0 | 91.5 | 95.8 | 97.0 | 95.5 | 90.3 | 93.6 |
| | DyGait Wang et al. (2023b) | ICCV'23 | 94.5 | 96.9 | 97.4 | 96.1 | 95.4 | 94.0 | 94.8 | **97.6** | **98.5** | **97.7** | 94.9 | 96.2 |
| | GaitBase Fan et al. (2023b) | CVPR'23 | - | - | - | - | - | - | - | - | - | - | - | 94.0 |
| | DANet Ma et al. (2023) | CVPR'23 | 95.0 | **97.3** | **98.3** | **97.4** | 94.7 | 91.0 | 93.9 | 97.4 | 98.2 | 97.6 | 94.2 | 95.9 |
| | GaitGCI Dou et al. (2023) | CVPR'23 | - | - | - | - | - | - | - | - | - | - | - | 96.6 |
| | QAGait Wang et al. (2024) | AAAI'24 | - | - | - | - | - | - | - | - | - | - | - | 94.6 |
| | CLASH Dou et al. (2024) | TIP'24 | - | - | - | - | - | - | - | - | - | - | - | 95.3 |
| | CLTD Xiong et al. (2024) | ECCV'24 | - | - | - | - | - | - | - | - | - | - | - | 96.4 |
| | LGDiffGait (Ours) | - | **95.8** | 97.2 | 97.9 | 96.8 | **96.8** | **94.3** | **95.5** | 96.8 | 98.4 | 97.3 | **96.4** | 96.7 |
| CL | GaitSet Chao et al. (2019) | AAAI'19 | 61.4 | 75.4 | 80.7 | 77.3 | 72.1 | 70.1 | 71.5 | 73.5 | 73.5 | 68.4 | 50.0 | 70.4 |
| | GaitPart Fan et al. (2020) | CVPR'20 | 70.7 | 85.5 | 86.9 | 83.3 | 77.1 | 72.5 | 76.9 | 82.2 | 83.8 | 80.2 | 66.5 | 78.7 |
| | GaitGL Lin et al. (2021) | ICCV'21 | 76.6 | 90.0 | 90.3 | 87.1 | 84.5 | 79.0 | 84.1 | 87.0 | 87.3 | 84.4 | 69.5 | 83.6 |
| | CSTL Huang et al. (2021) | ICCV'21 | 78.1 | 89.4 | 91.6 | 86.6 | 82.1 | 79.9 | 81.8 | 86.3 | 88.7 | 86.6 | 75.3 | 84.2 |
| | DyGait Wang et al. (2023b) | ICCV'23 | 82.2 | 93.0 | 95.2 | 91.6 | 87.1 | 83.4 | 87.2 | 90.1 | 92.4 | 88.2 | 75.8 | 87.8 |
| | GaitBase Fan et al. (2023b) | CVPR'23 | - | - | - | - | - | - | - | - | - | - | - | 77.4 |
| | DANet Ma et al. (2023) | CVPR'23 | 82.8 | **94.8** | 96.9 | **94.3** | **89.0** | 83.9 | 87.9 | **92.3** | **95.1** | **92.0** | 80.3 | 89.9 |
| | GaitGCI Dou et al. (2023) | CVPR'23 | - | - | - | - | - | - | - | - | - | - | - | 88.5 |
| | QAGait Wang et al. (2024) | AAAI'24 | - | - | - | - | - | - | - | - | - | - | - | 78.2 |
| | CLASH Dou et al. (2024) | TIP'24 | - | - | - | - | - | - | - | - | - | - | - | 89.3 |
| | CLTD Xiong et al. (2024) | ECCV'24 | - | - | - | - | - | - | - | - | - | - | - | 89.3 |
| | LGDiffGait (Ours) | - | **85.5** | 93.3 | **97.2** | 92.7 | 87.9 | **85.8** | **91.6** | 91.5 | 94.2 | 90.9 | **81.5** | 90.2 |

Table 2: Rank-1 accuracy (%) on OU-MVLP Takemura et al. (2018), excluding the identical-views cases.

| Method | Venue | Probe View | | | | | | | | | | | | | | Mean |
|---|---|---|---|---|---|---|---|---|---|---|---|---|---|---|---|---|
| | | 0° | 15° | 30° | 45° | 60° | 75° | 90° | 180° | 195° | 210° | 225° | 240° | 255° | 270° | |
| GaitSet Chao et al. (2019) | AAAI'19 | 79.5 | 87.9 | 89.9 | 90.2 | 88.1 | 88.7 | 87.8 | 81.7 | 86.7 | 89.0 | 89.3 | 87.2 | 87.8 | 86.2 | 87.1 |
| GaitPart Fan et al. (2020) | CVPR'20 | 82.6 | 88.9 | 90.8 | 91.0 | 89.7 | 89.9 | 89.5 | 85.2 | 88.1 | 90.0 | 90.1 | 89.0 | 89.1 | 88.2 | 88.7 |
| GaitGL Lin et al. (2021) | ICCV'21 | 84.9 | 90.2 | 91.1 | 91.5 | 91.1 | 90.8 | 90.3 | 88.5 | 88.6 | 90.3 | 90.4 | 89.6 | 89.5 | 88.8 | 89.7 |
| CSTL Huang et al. (2021) | ICCV'21 | 87.1 | 91.0 | 91.5 | 91.8 | 90.6 | 90.8 | 90.6 | 89.4 | 90.2 | 90.5 | 90.7 | 89.8 | 90.0 | 89.4 | 90.2 |
| GaitBase Fan et al. (2023b) | CVPR'23 | - | - | - | - | - | - | - | - | - | - | - | - | - | - | 90.0 |
| DANet Ma et al. (2023) | CVPR'23 | 87.7 | 91.3 | 91.6 | 91.8 | 91.7 | 91.4 | 91.1 | 90.4 | 90.3 | 90.7 | 90.9 | 90.5 | 90.3 | 89.9 | 90.7 |
| GaitGCI Dou et al. (2023) | CVPR'23 | 91.2 | 92.3 | 92.6 | **92.7** | 93.0 | 92.3 | 92.1 | 92.0 | 91.8 | 91.9 | 92.6 | 92.3 | 91.4 | 91.6 | 92.1 |
| DeepGaitV2 Fan et al. (2023a) | ARXIV'23 | - | - | - | - | - | - | - | - | - | - | - | - | - | - | 92.0 |
| CLASH Dou et al. (2024) | TIP'24 | 91.0 | 92.2 | 92.3 | 92.6 | 92.7 | 92.0 | 92.0 | 91.8 | 91.6 | 91.6 | 92.5 | 92.1 | 91.2 | 91.3 | 91.9 |
| CLTD Xiong et al. (2024) | ECCV'24 | **91.6** | **92.5** | **92.7** | 92.6 | **93.2** | **92.4** | 92.4 | 92.5 | 91.8 | 92.2 | 91.9 | **92.5** | 91.9 | 91.8 | **92.3** |
| LGDiffGait (Ours) | - | 90.8 | **92.5** | **92.7** | 92.5 | 92.7 | 92.1 | 92.0 | **92.8** | **92.7** | **92.6** | **92.7** | 91.7 | **91.8** | **92.3** | **92.3** |

## 4.2 COMPARISON WITH OTHER METHODS

We have compared the proposed LGDiffGait with various silhouette-based approaches, including set-based methods (GaitSet Chao et al. (2019) and GaitBase Fan et al. (2023b)), temporal modeling approaches (CSTL Huang et al. (2021), DyGait Wang et al. (2023b), and DANet Ma et al. (2023)), conv3D methods (GaitPart Fan et al. (2020), GaitGL Lin et al. (2021), and DeepGaitv2 Fan et al. (2023a)), and other SOTA models (GaitGCI Dou et al. (2023), HybridGait Dong et al. (2024), QAGait Wang et al. (2024), CLASH Dou et al. (2024), and CLTD Xiong et al. (2024)).

### 4.3 IMPLEMENTATION DETAILS

We mainly follow the experimental settings of DeepGaitV2 Fan et al. (2023a). Specifically, the silhouettes are resized to $64 \times 44$. Stochastic Gradient Descent (SGD) with an initial learning rate of 0.1 and a weight decay of 0.0005 is used as the optimizer. Data augmentations are similar to those in Fan et al. (2023b), including horizontal flipping, random erasing, rotation, perspective transformation, and affine transformation. The base channel $C$ in Figure 2 is set to 64. For the CASIA-B dataset, the batch size is set to (8, 16). The learning rate decays by 0.1 at 20k, 40k, and 60k iterations, with the models trained for a total of 80k iterations. For the OU-MVLP dataset, the batch size is set to (32, 8), with learning rate milestones at 60k, 80k, and 100k iterations, and the models trained for 120k iterations. For the GREW and Gait3D datasets, the batch size is set to (32, 4). The total iterations of GREW and Gait3D are 200k and 160k with milestones of (80k, 120k, 160k) and (40k, 80k, 120k), respectively. The number of blocks in the four layers is set to [1, 1, 1, 1] for CASIA-B and OU-MVLP, and [1, 3, 2, 1] for GREW and Gait3D. All experiments are conducted on $8\times$ NVIDIA RTX 4090 GPUs and are implemented using Pytorch Paszke et al. (2019) and OpenGait Fan et al. (2023b).

#### 4.3.1 RESULTS ON CASIA-B

Table 1 demonstrates that LGDiffGait achieves the highest accuracy across all conditions on the CASIA-B dataset. (1) Compared to set-based methods like GaitSet, LGDiffGait shows notable improvement, particularly under challenging conditions, with a 9.5% increase in BG and 19.8% in CL. This highlights the advantage of leveraging temporal dynamics rather than treating frames independently. (2) When compared to methods like GaitGL that use basic 3D CNNs for spatial-temporal modeling, LGDiffGait achieves superior results by capturing more detailed motion patterns. For instance, in the BG condition, LGDiffGait surpasses GaitGL by 2.2%, showing the effectiveness of combining local and global differences in temporal modeling. (3) Compared to other temporal modeling methods such as DyGait and DANet, LGDiffGait continues to perform better. Notably, in the CL condition, LGDiffGait achieves a Rank-1 accuracy of 90.2%, outperforming DyGait by 2.4% and DANet by 0.3%. These results underscore the robustness and effectiveness of LGDiffGait's local and global difference learning in capturing subtle motion variations, particularly in complex scenarios like cloth changing.

#### 4.3.2 RESULTS ON OUMVLP

Table 2 reports the performance of LGDiffGait on the OUMVLP dataset, demonstrating its effectiveness across various cross-view conditions. (1) LGDiffGait consistently achieves high accuracy across all viewpoints, with an average Rank-1 accuracy of 92.3%. The results indicate that our approach effectively captures representative and stable motion patterns, even in large-scale datasets. (2) Particularly challenging views, such as $0°$ and $180°$, typically yield lower accuracies due to less distinctive gait information. However, LGDiffGait manages to maintain robust performance, with a 90.8% accuracy at $0°$ and 92.8% at $180°$, reflecting its ability to mitigate the impact of confounding factors. (3) It is also noteworthy that, despite slight fluctuations in performance across certain angles, LGDiffGait demonstrates strong generalization, maintaining consistently high accuracy across the dataset, which underscores its potential for practical applications in gait recognition tasks where stability across diverse conditions is crucial.

#### 4.3.3 RESULTS ON GREW

Table 3 presents the performance comparison of various gait recognition methods on the GREW dataset. LGDiffGait achieves the highest accuracy across all evaluated metrics, with a Rank-1 accuracy of 82.7%, significantly outperforming other methods such as DeepGaitV2 and CLTD by 3.3% and 4.7%, respectively. This improvement suggests that LGDiffGait is highly effective in capturing discriminative gait features even in uncontrolled, real-world environments.

#### 4.3.4 RESULTS ON GAIT3D

The performance of various methods on the Gait3D dataset is summarized in Table 4, which evaluates methods under the challenging conditions of an unconstrained indoor environment. LGDiffGait

Table 3: Rank-1 accuracy (%), Rank-5 accuracy (%), Rank-10 accuracy (%), and Rank-20 accuracy (%) on the GREW Zhu et al. (2021) dataset.

| Method | Venue | Rank-1 | Rank-5 | Rank-10 | Rank-20 |
|---|---|---|---|---|---|
| GaitSet Chao et al. (2019) | AAAI'19 | 46.3 | 63.6 | 70.3 | 76.8 |
| GaitPart Fan et al. (2020) | CVPR'20 | 44.0 | 60.7 | 67.3 | 73.5 |
| GaitGL Lin et al. (2021) | ICCV'21 | 47.3 | 63.6 | 69.3 | 74.2 |
| CSTL Huang et al. (2021) | ICCV'21 | 50.6 | 65.9 | 71.9 | 76.9 |
| DyGait Wang et al. (2023b) | ICCV'23 | 71.4 | 83.2 | 86.8 | 89.5 |
| GaitBase Fan et al. (2023b) | CVPR'23 | 60.1 | - | - | - |
| GaitGCI Dou et al. (2023) | CVPR'23 | 68.5 | 80.8 | 84.9 | 87.7 |
| DeepGaitV2 Fan et al. (2023a) | ARXIV'23 | 79.4 | 88.9 | 91.4 | - |
| QAGait Wang et al. (2024) | AAAI'24 | 59.1 | 74.0 | 79.2 | 83.1 |
| CLASH Dou et al. (2024) | TIP'24 | 67.0 | 78.9 | 83.0 | 85.8 |
| CLTD Xiong et al. (2024) | ECCV'24 | 78.0 | 87.8 | - | - |
| LGDiffGait (Ours) | - | **82.7** | **91.2** | **93.7** | **95.6** |

demonstrates exceptional performance with a Rank-1 accuracy of 74.2%, surpassing the next best method, DeepGaitV2, by 1.4%. This indicates that LGDiffGait is particularly adept at handling the complexities of indoor gait recognition, where disturbing factors such as occlusion and misalignment are prevalent.

Table 4: Rank-1 accuracy (%), Rank-5 accuracy (%), mAP (%), and mINP on the Gait3D Zheng et al. (2022) dataset.

| Method | Venue | Rank-1 | Rank-5 | mAP | mINP |
|---|---|---|---|---|---|
| GaitSet Chao et al. (2019) | AAAI'19 | 36.7 | 58.3 | 30.0 | 17.3 |
| GaitPart Fan et al. (2020) | CVPR'20 | 28.2 | 47.6 | 21.6 | 12.4 |
| GaitGL Lin et al. (2021) | ICCV'21 | 29.7 | 48.5 | 22.3 | 13.3 |
| CSTL Huang et al. (2021) | ICCV'21 | 11.7 | 19.2 | 5.6 | 2.6 |
| DyGait Wang et al. (2023b) | ICCV'23 | 66.3 | 80.8 | 56.4 | 37.3 |
| GaitBase Fan et al. (2023b) | CVPR'23 | 65.6 | - | - | - |
| DANet Ma et al. (2023) | CVPR'23 | 48.0 | 69.7 | - | - |
| GaitGCI Dou et al. (2023) | CVPR'23 | 50.3 | 68.5 | 39.5 | 24.3 |
| DeepGaitV2 Fan et al. (2023a) | ARXIV'23 | 72.8 | 86.2 | 63.9 | - |
| HybridGait Dong et al. (2024) | AAAI'24 | 53.3 | 72.0 | 43.3 | 26.7 |
| QAGait Wang et al. (2024) | AAAI'24 | 67.0 | 81.5 | 56.5 | - |
| CLASH Dou et al. (2024) | TIP'24 | 52.4 | 69.2 | 40.2 | 24.9 |
| CLTD Xiong et al. (2024) | ECCV'24 | 69.7 | 85.2 | - | - |
| LGDiffGait (Ours) | - | **74.2** | **89.3** | **65.7** | **48.6** |

## 4.4 ABLATION STUDY

In this section, we conduct an ablation study on LGDiffGait to understand more clearly the contributions of different components within our proposed framework. All experiments in this study are performed on the Gait3D dataset, excluding the identical-views cases.

Table 5: Ablation study results on Gait3D Zheng et al. (2022) with different combinations of our proposed modules in LGDiff block.

| LDM | GDM | TAM | Rank-1 | Rank-5 | mAP | mINP |
|---|---|---|---|---|---|---|
| × | × | × | 71.2 | 85.7 | 62.9 | 44.3 |
| √ | × | × | 72.9 | 88.0 | 64.2 | 46.8 |
| × | √ | × | 72.6 | 87.6 | 63.4 | 46.3 |
| √ | √ | × | 73.8 | 88.6 | 64.8 | 47.1 |
| √ | √ | √ | **74.2** | **89.3** | **65.7** | **48.6** |

### 4.4.1 DIFFERENCE LEARNING MODULES

To evaluate the effectiveness of the Local Difference Module (LDM) and Global Difference Module (GDM) within the LGDiff block, we have conducted several ablation experiments as shown in Table 5. (1) When neither LDM nor GDM is applied, the baseline model achieves a Rank-1 accuracy of 71.2% on the Gait3D dataset. (2) Introducing the LDM alone improves Rank-1 accuracy to 72.9%, highlighting the importance of capturing local temporal differences in enhancing gait recognition performance. Notably, this improvement is more substantial than that achieved by the GDM alone, which increases Rank-1 accuracy to 72.6%. This suggests that local motion patterns, as captured by the LDM, are more critical for distinguishing fine-grained gait features than the broader global patterns captured by the GDM. (3) When both LDM and GDM are combined, the model achieves a Rank-1 accuracy of 73.8%, indicating that these two modules complement each other by capturing both fine-grained local differences and broader global motion patterns, thereby improving the overall recognition accuracy.

### 4.4.2 TEMPORAL ALIGNMENT MODULE

The impact of the Temporal Alignment Module (TAM) is also explored in the ablation study. As shown in Table 5, when TAM is not included, the model achieves a Rank-1 accuracy of 73.8%. However, when TAM is integrated along with both LDM and GDM, the Rank-1 accuracy increases to 74.2%, and mAP and mINP metrics also see notable improvements. These results demonstrate that TAM plays a crucial role in ensuring the temporal consistency of the extracted features, aligning local and global differences with the overall sequence dynamics, and thereby enhancing the discriminative power of the learned representations.

## 5 CONLUSION

In this paper, we present LGDiffGait, a novel framework for gait recognition that integrates both local and global difference modeling to capture subtle and comprehensive gait patterns. By incorporating the Local Difference Module (LDM) and Global Difference Module (GDM), our approach effectively highlights both short-range and long-range temporal variations in gait sequences. The Temporal Alignment Module (TAM) ensures that these difference features are seamlessly integrated with the main features, enhancing the overall discriminative power of the model. Extensive experiments conducted on several benchmark gait datasets, including CASIA-B, OUMVLP, GREW, and Gait3D, demonstrate the superior performance of LGDiffGait in comparison to state-of-the-art methods. While LGDiffGait achieves promising performance, the inclusion of LDM and GDM increases the network complexity. Future work will focus on optimizing these modules to reduce computational cost while maintaining accuracy.

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

## A  APPENDIX

### A.1  VISUALIZATION ANALYSIS

To further demonstrate the effectiveness of LGDiffGait, we provide visualization analyses using t-SNE Van der Maaten & Hinton (2008) for feature representation and Grad-CAM Selvaraju et al. (2017) for attention mapping. These visualizations underscore the model's ability to capture and emphasize salient gait features, aligning with the superior performance observed in our experiments.

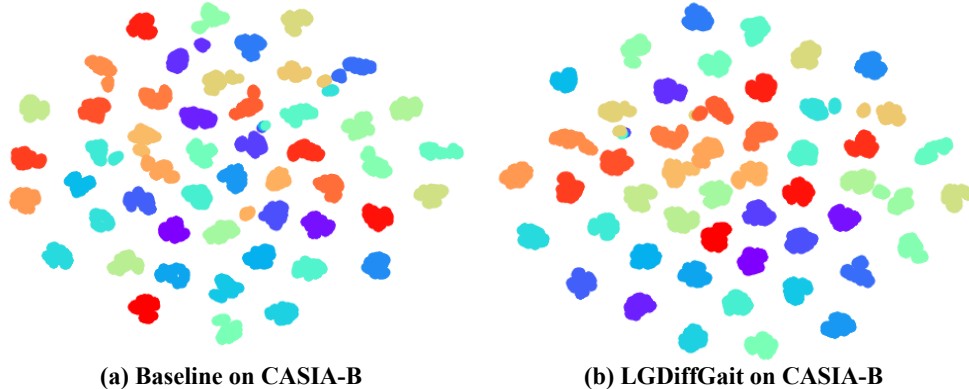

(a) **Baseline on CASIA-B**          (b) **LGDiffGait on CASIA-B**

Figure 5: Feature space visualization on CAISIA-B Yu et al. (2006) using t-SNE Van der Maaten & Hinton (2008). The baseline model refers to plain 3D CNNs.

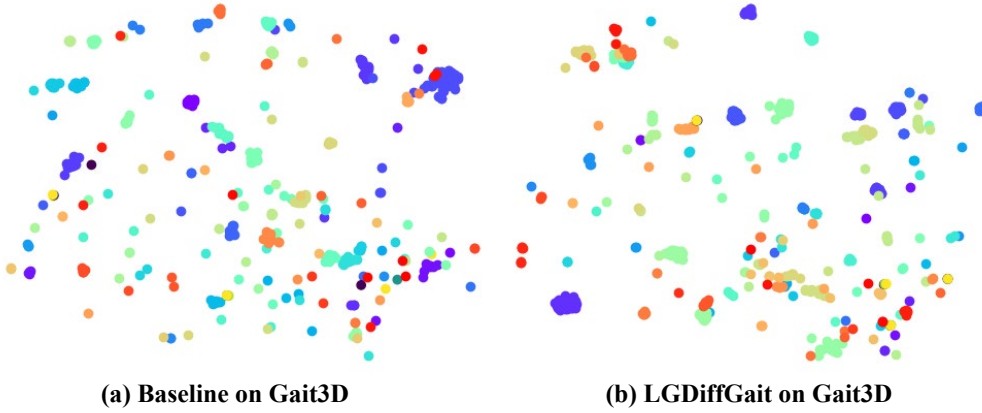

(a) **Baseline on Gait3D**          (b) **LGDiffGait on Gait3D**

Figure 6: Feature space visualization on Gait3D Zheng et al. (2022) using t-SNE Van der Maaten & Hinton (2008). The baseline model refers to plain 3D CNNs without difference modeling.

Figure 5 and Figure 6 present t-SNE visualizations on the CASIA-B and Gait3D datasets, respectively. Compared to the baseline 3D CNN model, LGDiffGait achieves more compact intra-class clustering and clearer inter-class separation, reflecting its enhanced ability to distinguish between different gait patterns. This improvement can be attributed to the local and global difference modeling modules, which capture subtle yet critical variations in gait, even in complex scenarios.

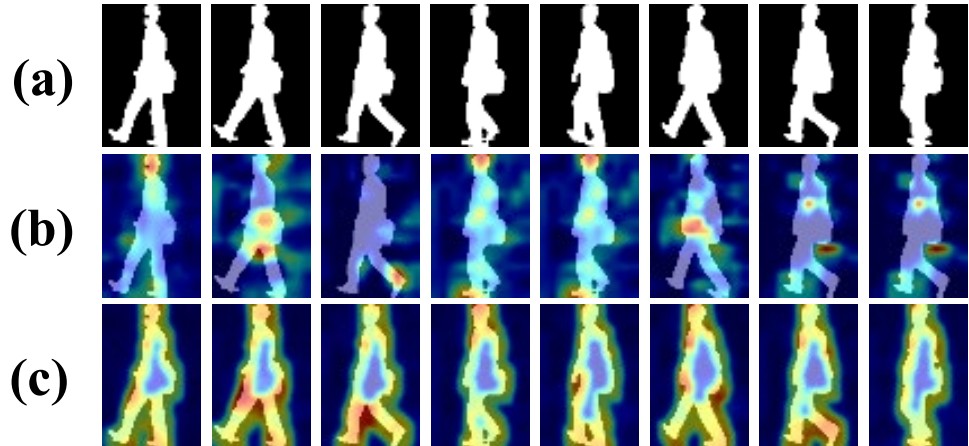

Figure 7: Grad-CAM Selvaraju et al. (2017) visualization on CASIA-B Yu et al. (2006). (a) shows input silhouette gait sequence. (b) presents the attention maps of baseline model. (c) depicts the attention maps of LGDiffGait.

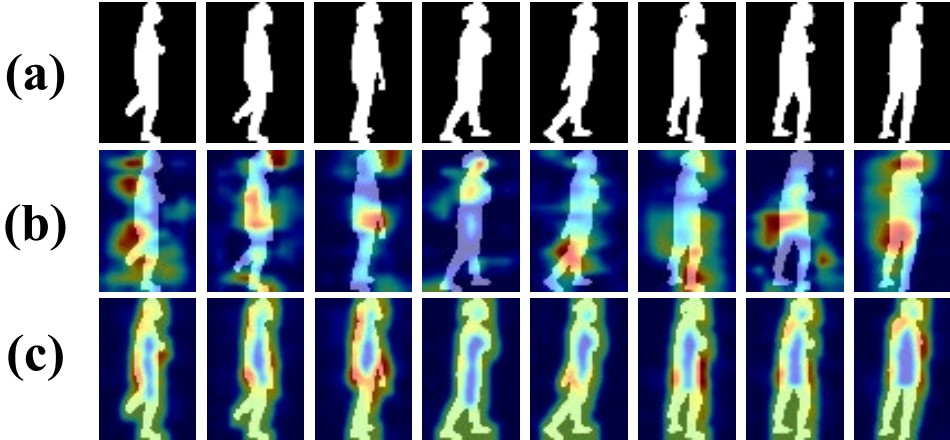

Figure 8: Grad-CAM Selvaraju et al. (2017) visualization on Gait3D Zheng et al. (2022).

Figure 7 and Figure 8 showcase Grad-CAM visualizations comparing the baseline model to LGDiffGait on the same datasets. The attention maps generated by LGDiffGait focus more effectively on key regions, particularly the dynamic movements of the legs and arms. These areas are essential for accurate gait recognition, and LGDiffGait's precise attention to these regions improves both the interpretability and robustness of its predictions.

In summary, the t-SNE results demonstrate LGDiffGait's superior feature extraction, leading to improved class separability, while the Grad-CAM heatmaps illustrate its ability to focus on the most relevant areas of the gait sequence. Together, these visualizations provide compelling evidence of LGDiffGait's effectiveness in learning and utilizing critical gait features.

