# OpenReview forum: "LGDiffGait: Local and Global Difference Learning for Gait Recognition with Silhouettes"
_ICLR.cc/2025/Conference — Submitted to ICLR 2025_

### Official Review · Reviewer_9NpQ · 2024-10-29

**Soundness:** 3
**Presentation:** 2
**Contribution:** 2
**Rating:** 5
**Confidence:** 5

**Summary:**

The paper introduces LGDiffGait, a framework for gait recognition that utilizes Local and Global Difference Modules (LDM and GDM) to capture fine-grained and broad temporal features from silhouettes. A Temporal Alignment Module (TAM) further aligns these features across sequences, resulting in state-of-the-art performance on multiple benchmark datasets.

**Strengths:**

- LGDiffGait employs a dual-level differentiation approach with a Temporal Alignment Module (TAM) that captures both subtle and broad temporal features, ensuring cohesive alignment across sequences.

- Sufficient Comparisons.

- Visualizations using t-SNE and Grad-CAM effectively illustrate the model’s attention to key regions, particularly in capturing dynamic limb movements, which enhances interpretability.

**Weaknesses:**

- **Broader Comparison with Temporal Methods**: Expanding comparisons with other temporal methods would better contextualize LGDiffGait’s specific advantages, situating it within the broader landscape of temporal gait recognition models.

- **Validation of Temporal Method Generalizability**: To fully assess the generalizability of the temporal methods (LDM, GDM, and TAM), applying these modules to various baseline models (e.g., GaitBase and DeepGaitV2) would provide a clearer demonstration of their adaptability and effectiveness across different architectures.

- **Lack of Efficiency Metrics**: The absence of parameter and FLOP metrics limits understanding of the model’s computational demands, which would be valuable for assessing its scalability and efficiency.

- **Poor Novelty** The community may find it hard to get some ideas new from the manuscript. The local and global shifted (diff) temporal modeling has been discussed many times in previous works[1, 2, 3]. The authors have made much effort in this topic by still have not achieved impressive enough performance improvements among all the employed datasets.

[1] Lin et al, GaitGL at ICCV2021
[2] Lin et al, MT3D at MM2020
[3] Zheng et al, MSTGait at MM2023

**Questions:**

Addressing issues related to novelty, generalizability, efficiency metrics, and broader comparative analysis would further strengthen this paper's impact.

---

### Official Review · Reviewer_yZK7 · 2024-11-02

**Soundness:** 3
**Presentation:** 3
**Contribution:** 2
**Rating:** 3
**Confidence:** 5

**Summary:**

The authors found the problem of existing methods that mainly focus on extracting the feature on the entire gait sequence, so they introduced the LGDiff block to get the difference in local and global levels with a temporal alignment module to help the model focus on more detailed movement. Based on the experimental results, the performance over four datasets is higher than the SOTA methods

**Strengths:**

1. The paper is easy to understand.
2.  The figures are clear and the tables are easy to read.
3. The proposed method is reasonable in that more hand-craft features are involved in the feature extraction leading the overall performance improvement.

**Weaknesses:**

1. How does the alignment module align temporal information? It just combines the main and the difference features, and it is not proper to define it as 'align'
	2. How much the model size is increased? It seems it introduces a dual network to extract the difference features. And DeepGaitV2 is big enough, LGDiffGait is likely to be a larger model, so it is hard to say the improvement is solely from a nice model design or better feature
	3. The authors said the difference is an essential feature to measure the detailed movement. Do you try to use the difference only to see how it performs?  The idea is similar to using the optical flow to describe the motion.
         4. The improvement of performance does not represent extract nuance, it may be due to overfitting on some non-gait-related objects. It is better to use an attention map or cross-domain evaluation to show the effectiveness.
        5. There is a lack of analysis about why this design is good and why it works well.

**Questions:**

Optical flow also focuses on the difference, so what is the advantage of using this LGDiff?
Since the difference could capture the nuances, why does the model need the main convolution branch rather than just using the difference?

---

### Official Review · Reviewer_YQvo · 2024-11-02

**Soundness:** 2
**Presentation:** 3
**Contribution:** 2
**Rating:** 3
**Confidence:** 5

**Summary:**

This paper presents a gait recognition framework named LGDDiffGait, which incorporates Local and Global Difference (LGDiff) blocks. The LGDiff block consists of two components: a Local Difference Module (LDM) and a Global Difference Module (GDM). The LDM captures local motions between adjacent frames within a sliding window, while the GDM captures global differences across the whole sequence. A Temporal Alignment Module (TAM) is further used to align the extracted local and global differences with the overall sequence dynamics. Experiments on four gait datasets demonstrate that the proposed method achieves SOTA gait recognition performance.

**Strengths:**

+ Using the difference information along the temporal dimension is reasonable for enhancing gait recognition.
+ Experimental results show the SOTA performance of the proposed method.

**Weaknesses:**

- The biggest concern is the theoretical novelty of the proposed method. The use of difference features has already been explored in DyGait (Wang et al. 2023b), which is almost the same as the global difference module in this work. The primary distinction lies in the introduction of the local difference module, which shifts the extraction of difference features from the entire sequence—as utilized in the global difference module—to differences across several adjacent frames within a sliding window, which is a minor modification. In addition, the learning of local features has been widely applied in gait recognition, both in spatial and temporal domains, and is not a new concept. Consequently, these factors limit the technical contribution of this paper.

- The proposed framework and the approach to learning difference features are primarily tailored for the specific task of gait recognition, offering limited insights for broader tasks or other areas of representation learning. So, this paper may not be ideally suited for ICLR.

- In the temporal alignment module, it is explained that the difference features are aligned with the overall sequence dynamics. However, from my understanding, the temporal order is preserved when extracting difference features, so it is unclear why temporal misalignment of the difference features would occur. In addition, temporal alignment is proposed to be achieved by concatenating the main features with the difference features and further applying a 3D convolution. The rationale behind these operations for achieving temporal alignment also requires further clarification.

- There are a few typos in the paper; for example, in Section 3.2.4, deisned -> designed. Some descriptions of related works may not be entirely accurate. For example, SMPLGait (Zheng et al., 2022) is more accurately described as a fusion of appearance-based and model-based methods rather than purely model-based. The 3D model is used solely to learn view transformation for silhouette features, with recognition relying exclusively on silhouette features.

**Questions:**

In the first row of Table 5, does this indicate that the whole LGDDiff blocks have been removed? If that is the case, does the evaluated backbone contain only 3D conv + TP + HPP, yet still achieve such high performance?

---

### Official Review · Reviewer_Q7Nq · 2024-11-02

**Soundness:** 2
**Presentation:** 1
**Contribution:** 3
**Rating:** 5
**Confidence:** 3

**Summary:**

The paper presents an approach for gait identification called Local and Global Difference Learning for Gait Recognition with Silhouettes (LGDiffGait). The method incorporates local and global gait features in a unique representation. The approach is evaluated on different public datasets and, compared to existing methods, provides superior results.

**Strengths:**

- The paper considers an interesting problem, whose relevance goes beyond gait recognition
- The state-of-the-art is fairly discussed (with few exceptions, see Questions)
- The results are superior to existing approaches on different public datasets

**Weaknesses:**

- The presentation of the method and the procedure flow is not fully clear (see Questions). I think this is
- The limitations of the approach are only briefly touched (the authors mention the computational aspects)

**Questions:**

- About existing approaches:  the discussion does not mention works based on architectures for sequences (e.g. LSTMs or Transformers). Are these approaches missing? Can you discuss how your approach compares to them, if existing in literature?

- About the need for both local and global temporal representations of motion, existing deep architectures as the SlowFast (https://arxiv.org/abs/1812.03982) addressed this problem. From what I can gather, it seems to me that the attempt of the authors is different, in the sense they want to keep the model complexity under control. Nevertheless, a discussion in relation to these already existing approaches would be beneficial to fully appreciate the intentions of the authors and to better contextualize your design choices

- An important influence on the performance of the method is from the silhouettes in input. Comments in this sense are missing

- The reader is a bit lost in the details of the method. Although in some parts they are even redundant (e.g. when describing twice, with text and with a formula, the main architectural operations), in my opinion a clear storytelling of the method is missing. In particular, it is unclear to me the flow in the forward propagation. What's the input? A single image,  image pairs, the whole sequence? [Further doubts on this part are related to some of my questions below.]

- Related to the first point, I miss the meaning of Fig. 1. Should this be intended as an example of input? Under what circumnstances we are facing the different situations? I suggest you to provide a more detailed caption of the figure, clarifying the purpose of the figure.

- In sec. 3.2.2 the need for the padding is mentioned, but the technical/practical motivations are unclear

- It would be nice to have an intuition on the behavior of the Local Difference Module with an example (an image?)

- When computing the differencing steps, the procedure is reminiscent of a change detection approach. Is this correct?

- The index t appears only in the GDM, so it is not clear to me how the sequence is processed

- The presence of a triplet loss unveils that a specific training strategy is adopted, but this is introduced only in Sec. 3.3 with no appropriate discussion. How is the training organized? I suggest you to provide a more detailed explanation of the training procedure (including for instance the sampling strategies used for the input pairs) in the point of the paper you find the most appropriate.

- The results from all methods are very high in general, with no particular coherence between the different views or any common pattern as the viewing angle is changing. Any intuition on the reasons why? Can this give suggestions on the nature of the datasets or the generalization capabilities of the methods? What are the implications for the practical applicability of gait recognition systems?

- A thorough discussion on limitations would be appreciated

- A comment on ethical aspects is needed

**Details Of Ethics Concerns:**

The paper is about gait identification from videos. Only public data have been employed here, but some comments on ethical aspects may be beneficial.

---

### Official Review · Reviewer_X2dM · 2024-11-03

**Soundness:** 2
**Presentation:** 3
**Contribution:** 2
**Rating:** 3
**Confidence:** 5

**Summary:**

The paper introduces an approach to gait recognition that leverages both local and global difference learning within video silhouettes to enhance feature extraction. The method uses Local Difference Modules (LDM) and Global Difference Modules (GDM) to capture intricate motion details across both short and long temporal spans, with a Temporal Alignment Module ensuring consistency across the extracted features. The framework significantly outperforms existing methods on multiple benchmarks, demonstrating its robustness and effectiveness in gait recognition across diverse conditions.

**Strengths:**

+ Well-structured paper with clear explanations of the methods and results.

+ Demonstrates state-of-the-art results on multiple gait recognition datasets, showing improvements over existing methods.

**Weaknesses:**

1. The paper introduces the concept of local and global gait differences without a thorough discussion of the underlying motivations or theoretical foundations compared to traditional spatial-temporal approaches. Insightful exploration into specific scenarios where existing methods fail could substantiate the need for this new approach. A deeper analysis would help clarify why the proposed method better captures unique gait characteristics, potentially through comparative studies or by linking the approach to fundamental biomechanical principles of human motion.

2. Noise in silhouette data could affect difference accuracy. The reliance on pre-processed silhouette data, which is susceptible to noise from segmentation and alignment errors, raises concerns about the integrity of the gait differences captured by the model. This method's effectiveness might be compromised if these preprocessing steps introduce artifacts that are mistaken for intrinsic gait differences. The paper could benefit from a robust discussion on preprocessing techniques' reliability and strategies to mitigate their impact, ensuring that the gait differences reflect true biomechanical motion rather than processing inaccuracies.

3. Absence of cross-dataset evaluation limits the demonstrated generalizability of the LGDiffGait model. Including such evaluations would not only validate the model's robustness across varied settings but also highlight its performance stability amidst different capture conditions and demographic variabilities. Insights into how the model performs when trained on one dataset and tested on another could underscore its utility in real-world applications and help identify potential biases or limitations in dataset-specific training.

4. It would be advantageous for the research to examine the model's applicability to RGB data, which remains unexplored and thus limits its use in scenarios where only RGB data is available. It would be valuable to discuss or demonstrate how the model could be adapted for RGB inputs, potentially expanding its practical relevance and adoption. Exploring methodologies to integrate color and texture information available in RGB data could potentially enhance the model’s discriminatory power by leveraging additional cues beyond silhouette shapes.

5. It would be beneficial for the paper to explore the impact of frame step size on the performance of gait recognition. Since the frame interval can significantly influence the detection of subtle gait differences, investigating optimal step sizes for different gait speeds or conditions could yield deeper insights. It would be informative to analyze how varying intervals affect the model’s ability to detect meaningful differences, which would enhance our understanding of the model’s sensitivity and operational flexibility.

**Questions:**

1. What are the computational costs associated with the LGDiffGait model?

2. How does the LGDiffGait model handle noisy silhouette data resulting from poor segmentation or alignment processes during preprocessing?

3. Are there ongoing or planned future works to adapt the LGDiffGait framework for use with RGB data? What potential methodologies or modifications are being considered to incorporate color and texture information into the current model?

---

### Meta-Review · Area_Chair_FTaw · 2024-12-19

**Metareview:**

This paper introduces a gait recognition method that considers both local and global difference learning in video silhouettes to enhance feature extraction, namely, incorporates local and global gait features in a unique representation able to capture motion details across both short and long temporal spans, also adopting a temporal alignment mechanism to ensure consistency across the extracted features. The framework significantly outperforms existing methods on multiple benchmarks, demonstrating its robustness and effectiveness in gait recognition across diverse conditions.

Other than a general appreciation of this work, several negative aspects and points of discussion have been raised by the reviewers. They mainly deal with the not fully original concept introduced, the weak motivations, insufficient experimental analysis, ablations and comparative tests, poor explanation or justification of some steps of the approach proposed, somewhat unclear description of the approach, and missing discussion of the method's limitations.

The authors did not provide a rebuttal to these comments, hence this paper cannot be accepted for publication to ICLR 2025.

**Additional Comments On Reviewer Discussion:**

No rebuttal was provided by the authors.

---

### Decision · Program_Chairs · 2025-01-22

Reject